# Optimizing Polynomial Graph Filters: A Novel Adaptive Krylov Subspace Approach

## ABSTRACT

Graph Neural Networks (GNNs) as spectral graph filters, enhancing specific frequencies of graph signals while suppressing the rest, find a wide range of applications in web networks. To bypass eigendecomposition, polynomial graph filters are proposed to approximate graph filters by leveraging various polynomial bases for filter training. However, no existing studies have explored the diverse polynomial graph filters from a unified perspective for optimization.

In this paper, we first unify polynomial graph filters, as well as the optimal filters of *identical* degrees into the Krylov subspace of the *same* order, thus providing equivalent expressive power theoretically. Next, we investigate the asymptotic convergence property of polynomials from the unified Krylov subspace perspective, revealing their limited adaptability in graphs with varying heterophily degrees. Inspired by those facts, we design a novel adaptive Krylov subspace approach to optimize polynomial bases with provable controllability over the graph spectrum so as to adapt various heterophily graphs. Subsequently, we propose AdaptKry, an optimized polynomial graph filter utilizing bases from the adaptive Krylov subspaces. Meanwhile, in light of the diverse spectral properties of complex graphs comprising numerous components, we extend AdaptKry by leveraging multiple adaptive Krylov bases without incurring extra training costs. As a consequence, extended AdaptKry is able to capture the intricate characteristics of graphs and provide insights into their inherent complexity. We conduct extensive experiments across a series of real-world datasets. The experimental results demonstrate the superior filtering capability of AdaptKry, as well as the optimized efficacy of the adaptive Krylov basis.

## KEYWORDS

Spectral Graph Neural Networks, Supervised Classification, Krylov Subspace

**ACM Reference Format:**

Anonymous Author(s). 2024. Optimizing Polynomial Graph Filters: A Novel Adaptive Krylov Subspace Approach. In *Proceedings of Proceedings of the ACM Web Conference 2023 (WWW '24)*. ACM, New York, NY, USA, 12 pages.

## 1 INTRODUCTION

Graph Neural Networks (GNNs) [23], inherently acting as *spectral graph filters* [5, 10, 23, 42], are widely employed across various web

applications, such as classification on web services [16, 51], online e-commerce [11, 48, 50], and analysis of social networks [27, 35, 37]. Specifically, given a graph $\mathbf{G}$, the Laplacian matrix $\mathcal{L}$ is eigendecomposed as $\mathcal{L} = \mathbf{U}\mathbf{\Lambda}\mathbf{U}^\top$ where $\mathbf{U}$ is the matrix of eigenvectors and $\mathbf{\Lambda}$ is the diagonal matrix of eigenvalues. Given a signal $\mathbf{x}$, a spectral graph filter $g_{\mathbf{w}}$ parameterized by vector $\mathbf{w}$ on $\mathbf{x}$ is $g_{\mathbf{w}}(\mathcal{L})\mathbf{x} = \mathbf{U}g_{\mathbf{w}}(\mathbf{\Lambda})\mathbf{U}^\top\mathbf{x}$. To avert the computation burden of eigendecomposition, polynomial filters have been proposed to approximate the optimal graph filter via polynomial approximation, such as ChebNet [10], BernNet [21], and JacobiConv [44]. Specifically, ChebNet utilizes truncated Chebyshev polynomials [19, 30] and achieves localized spectral filtering. To acquire better controllability and interpretability, BernNet employs Bernstein polynomials [12]. Lately, Wang and Zhang [44] analyze the expressive power of current polynomial filters and propose JacobiConv by exploiting Jacobi polynomial bases [1].

Numerous studies have investigated various polynomials for polynomial filters, yet no prior research has explored the properties of polynomials as signal bases for graph filters from a unified perspective. Meanwhile, those existing polynomials are usually exploited directly in graph filters without any customization for different graphs. An intriguing and valuable question that arises is: *can we optimize polynomial graph filters by enhancing the underlying polynomial basis?*

To address this question, we revisit polynomial filters in terms of *Krylov subspace* [29, 32]. The Krylov subspace method was initially proposed in the early 1950s for solving linear systems [36, 43]. A Krylov subspace is a subspace of Euclidean vector space. In particular, an order-$K$ Krylov subspace is constructed by multiplying the first $K$ powers of a matrix $\mathbf{A} \in \mathbb{R}^{n \times n}$ to a vector $\mathbf{v} \in \mathbb{R}^n$ where $K$ is an integer (details in Section 6). In the realm of polynomial filters, matrix $\mathbf{A}$ corresponds to the propagation matrix, $\mathbf{v}$ is the feature signal, and $K$ stands for the propagation hops. In this paper, we demonstrate that polynomial filters sharing identical degrees inherently belong to the *Krylov subspace* of the same order and hence yield theoretically equivalent expressive power.

From the perspective of the Krylov subspace, we prove that propagation matrices in existing polynomials gradually approach stability in an asymptotic manner regardless of the heterophily in the underlying graphs (Theorem 1). It is known that high-frequency signals can also provide insightful information [2, 7, 25], especially for strong heterophily graphs. Inspired, we design a tunable propagation matrix and propose a novel adaptive Krylov subspace approach. To clarify, we introduce a hyperparameter into propagation matrices by inspecting the graph heat equation [9, 45], equipping the ability to reshape the underlying spectrum of graphs (Theorem 2). As a consequence, polynomials from the adaptive Krylov subspace disrupt the convergence pattern and offer flexible adaptability to graphs with varying heterophily. Subsequently, we propose AdaptKry, an optimized polynomial graph filter by leveraging bases

from the adaptive Krylov subspaces, known as *adaptive Krylov bases*. Meanwhile, complex graphs consisting of multiple components are inclined to exhibit diverse spectral properties. To handle such scenarios, we extend AdaptKry by employing several adaptive Krylov bases simultaneously. The resultant basis set also sheds light on the complexity of networks. In addition, we integrate weight parameters of different bases into one *singular* parameter, ensuring that no extra training overhead is incurred. We compare AdaptKry against 11 baselines on 6 real-world datasets with a range of homophily ratios. AdaptKry achieves the highest accuracy scores in node classification for almost all cases. Those experimental results strongly support the superior performance of AdaptKry as polynomial filters. We also devise a comprehensive ablation study to demonstrate the properties of adaptive Krylov bases and AdaptKry in Section 5.3. In a nutshell, our contributions are briefly summarized as follows.

- We unify polynomial filters as well as optimal filters from the Krylov subspace perspective and reveal their constrained adaptability in graphs with varying heterophily degrees.

- We devise a novel adaptive Krylov subspace approach and propose an optimized polynomial filter AdaptKry. Meanwhile, we extend AdaptKry to incorporate multiple adaptive Krylov bases, enhancing expressive capability without extra training costs.

- We conduct comprehensive experiments to verify the superior performance of AdaptKry as polynomial filters, as well as extensive ablation study to explore the properties AdaptKry and the adaptive Krylov basis.

## 2 PRELIMINARY

**Table 1: Frequently used notations**

| Notation | Description |
|---|---|
| $G = (\mathcal{V}, \mathcal{E})$ | a social network with node set $\mathcal{V}$ and edge set $\mathcal{E}$ |
| $n, m$ | the numbers of nodes and edges in $G$ respectively |
| $d_u$ | the degree of node $u$ |
| $\mathcal{N}_u, \mathcal{N}_u^{(\ell)}$ | the one-hop and $\ell$-th hop neighbors of node $u$ |
| $\mathbf{A}, \mathbf{D}$ | the adjacency matrix and diagonal degree matrix |
| $\mathcal{L}$ | the normalized Laplacian matrix |
| $\mathbf{X}, \mathbf{x}_u$ | the feature matrix and the feature vector of node $u$ |
| $\mathbf{Z}, \mathbf{z}_u$ | the representation matrix and the representation vector of node $u$ |
| $\mathcal{K}_K(\mathbf{P}, \mathbf{x})$ | A Krylov subspace constructed by matrix $\mathbf{P}$ and vector $\mathbf{x}$ as $\mathcal{K}_K(\mathbf{P}, \mathbf{x}) = \text{span}\{\mathbf{x}, \mathbf{Px}, \mathbf{P}^2\mathbf{x}, \cdots, \mathbf{P}^{K-1}\mathbf{x}\}$ |

### 2.1 Notations and Definitions

In this paper, we use bold uppercase letters, bold lowercase letters, and calligraphic fonts to represent matrices (e.g., $\mathbf{A}$), vectors (e.g., $\mathbf{x}$), and sets (e.g., $\mathcal{N}$), respectively. The $i$-th row (resp. column) of matrix $\mathbf{A}$ is represented by $\mathbf{A}[i, \cdot]$ (resp. $\mathbf{A}[\cdot, i]$). We denote $[n] = \{1, 2, \cdots, n\}$.

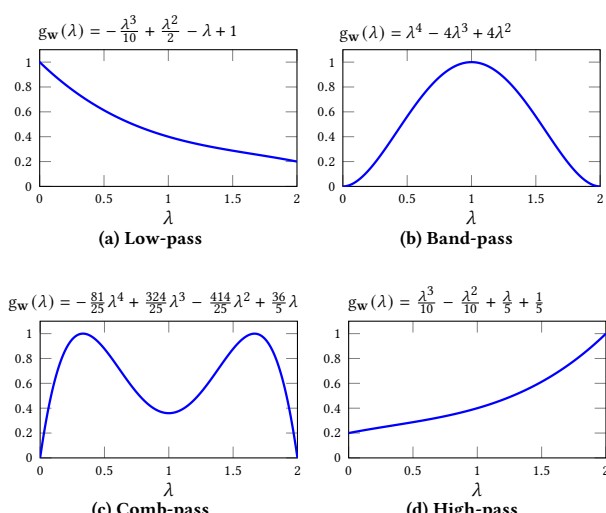

$$g_{\mathbf{w}}(\lambda) = -\frac{\lambda^3}{10} + \frac{\lambda^2}{2} - \lambda + 1$$
(a) Low-pass

$$g_{\mathbf{w}}(\lambda) = \lambda^4 - 4\lambda^3 + 4\lambda^2$$
(b) Band-pass

$$g_{\mathbf{w}}(\lambda) = -\frac{81}{25}\lambda^4 + \frac{324}{25}\lambda^3 - \frac{414}{25}\lambda^2 + \frac{36}{5}\lambda$$
(c) Comb-pass

$$g_{\mathbf{w}}(\lambda) = \frac{\lambda^3}{10} - \frac{\lambda^2}{10} + \frac{\lambda}{5} + \frac{1}{5}$$
(d) High-pass

**Figure 1: Illustrations of polynomial approximations to four common graph filters**

Let $G = (\mathcal{V}, \mathcal{E})$ be an undirected and connected graph with node set $\mathcal{V}$ and edge set $\mathcal{E}$, $n = |\mathcal{V}|$ and $m = |\mathcal{E}|$ be the number of nodes and edges, respectively. Let $\mathbf{X} \in \mathbb{R}^{n \times d}$ be its $d$-dimension feature matrix where each node $v \in \mathcal{V}$ is associated with a $d$-dimensional feature vector $\mathbf{x}_v \in \mathbf{X}$. For simplicity purposes, we also use node $u \in \mathcal{V}$ to indicate its index, i.e., $\mathbf{x}_u = \mathbf{X}[u, \cdot]$. The direct neighbor set of any node $u \in \mathcal{V}$ with degree $d_u$ is denoted as $\mathcal{N}_u$ with $d_u = |\mathcal{N}_u|$. Let $\mathbf{A} \in \mathbb{R}^{n \times n}$ be the adjacency matrix of $G$, i.e., $\mathbf{A}[u, v] = 1$ if $\langle u, v \rangle \in \mathcal{E}$; otherwise $\mathbf{A}[u, v] = 0$, and $\mathbf{D} \in \mathbb{R}^{n \times n}$ be the diagonal degree matrix of $G$, i.e., $\mathbf{D}[u, u] = d_u$. Normalized Laplacian matrix $\mathcal{L}$[1] of graph $G$ is defined as $\mathcal{L} = \mathbf{I} - \mathbf{D}^{-\frac{1}{2}}\mathbf{A}\mathbf{D}^{-\frac{1}{2}}$ where $\mathbf{I}$ is the identity matrix.

### 2.2 Polynomial Graph Filters

Given a graph $G = (\mathcal{V}, \mathcal{E})$, the corresponding Laplacian matrix is $\mathcal{L} = \mathbf{U}\boldsymbol{\Lambda}\mathbf{U}^{\top}$, where $\mathbf{U}$ is the matrix of eigenvectors and $\boldsymbol{\Lambda} = \text{diag}[\lambda_1, \cdots, \lambda_n]$ is the diagonal matrix of eigenvalues. In particular, eigenvalues $\lambda_i \in [0, 2]$ for $i \in [n]$ mark the *frequency* and the eigenvalue set $\{\lambda_1, \cdots, \lambda_n\}$ is the *spectrum* of $G$. Without loss of generality, we assume $0 = \lambda_1 \leq \lambda_2 \leq \cdots \leq \lambda_n \leq 2$.

Given a graph signal $\mathbf{x} \in \mathbb{R}^n$, graph Fourier operator $\mathcal{F}(\mathbf{x}) = \mathbf{U}^{\top}\mathbf{x}$ transforms the graph signal $\mathbf{x}$ into the spectral domain, and then a predefined spectral filtering $g_{\mathbf{w}}(\cdot)$ parameterized by $\mathbf{w} \in \mathbb{R}^n$ is applied on the transformed signals. Last, the processed signal is transformed back via the inverse graph Fourier transform operator $\mathcal{F}^{-1}(\mathbf{x}) = \mathbf{Ux}$. Specifically, the spectral graph filter is defined as

$$\mathcal{F}^{-1}(\mathcal{F}(g_{\mathbf{w}}) \odot \mathcal{F}(\mathbf{x})) = \mathbf{U}g_{\mathbf{w}}(\boldsymbol{\Lambda})\mathbf{U}^{\top}\mathbf{x}$$
$$= \mathbf{U} \, \text{diag}(g_{\mathbf{w}}(\lambda_1), \cdots, g_{\mathbf{w}}(\lambda_n))\mathbf{U}^{\top}\mathbf{x}. \quad (1)$$

where $\odot$ is the Hadamard product.

In general, the filter function $g_{\mathbf{w}}(\cdot)$ enhances signals in specific frequency intervals and suppresses the rest parts. Four types of

---
[1]Without further specification, $\mathcal{L}$ refers to the *normalized* Laplacian matrix in this paper.

spectral filters are commonly observed in real-world graphs [21], i.e., low-pass filter, band-pass filter, comb-pass filter, and high-pass filter, as illustrated in Figure 1. Intuitively, homophily graphs tend to contain low-frequency signals whilst heterophily graphs likely own high-frequency signals. To quantify the graph homophily degree, we define *homophily ratio* $h$ as follows.

DEFINITION 1 (HOMOPHILY RATIO $h$). *Given a graph* $\mathbf{G} = (\mathcal{V}, \mathcal{E})$ *and its label set* $C \in \mathbb{R}^{|\mathcal{V}|}$, *the homophily ration* $h$ *of* $\mathbf{G}$ *is the fraction of edges with two end nodes from the same class, i.e.,* $h = \frac{|\{\langle u,v \rangle \in \mathcal{E}: C_u = C_v\}|}{|\mathcal{E}|}$.

Since (i) it is inefficient to conduct matrix eigendecomposition (time complexity $O(n^3)$) and (ii) any desired spectral filters can be approximated with polynomials [19, 30], existing spectral GNNs approximate the spectral filter with various learnable polynomials.

**ChebNet.** Defferrard et al. [10] adopt Chebyshev polynomials up to $K$-th order to approximate Equation (1) as

$$\mathbf{U}\mathrm{g}_{\mathbf{w}}(\mathbf{\Lambda})\mathbf{U}^\top \mathbf{x} \approx \mathbf{U}\left(\sum_{k=0}^{K} w_k T_k(\hat{\mathbf{\Lambda}})\right)\mathbf{U}^\top \mathbf{x} = \left(\sum_{k=0}^{K} w_k T_k(\hat{\mathcal{L}})\right)\mathbf{x}, \quad (2)$$

where $\hat{\mathcal{L}} = \mathbf{U}\hat{\mathbf{\Lambda}}\mathbf{U}^\top$ and $\hat{\mathbf{\Lambda}} = 2\mathbf{\Lambda}/\lambda_{\max} - \mathbf{I}$ is a normalized diagonal matrix with eigenvalues within $[-1, 1]$, and $\lambda_{\max} = \max\{\lambda_1, \cdots, \lambda_n\}$ is the maximum eigenvalue of $\mathcal{L}$. $w_k$ is the learnable Chebyshev coefficient and $T_k(x)$ is the Chebyshev polynomial recursively computed as $T_k(x) = 2x T_{k-1}(x) - T_{k-2}(x)$ with $T_0(x) = 1$ and $T_1(x) = x$. By taking the learnable Chebyshev coefficient $w_k$ as the trainable parameter of the $k$-th convolution layer, the filter function of ChebNet is $\mathrm{g}_{\mathbf{w}}(\lambda) = \sum_{k=0}^{K} w_k T_k(\lambda)$.

**GPR-GNN.** Chien et al. [8] follow the generalized PageRank (GPR) architecture by stacking up $K$ convolution layers and assigning a learnable weight for each layer. They propose GPR-GNN as

$$\mathbf{z} = \sum_{k=0}^{K} w_k (\tilde{\mathbf{D}}^{-\frac{1}{2}}\tilde{\mathbf{A}}\tilde{\mathbf{D}}^{-\frac{1}{2}})^k \cdot f_\theta(\mathbf{x}), \quad (3)$$

where $\tilde{\mathbf{A}} = \mathbf{A} + \mathbf{I}$, $\tilde{\mathbf{D}} = \mathbf{D} + \mathbf{I}$, $\mathbf{z}$ is the final node representation, $f_\theta(\cdot)$ represents a neural network parameterized by $\theta \in \mathbb{R}^n$, and $w_k$ is the learnable parameter of $k$-th convolutional layer. Hence, the filter function of GPR-GNN is $\mathrm{g}_{\mathbf{w}}(\tilde{\lambda}) = \sum_{k=0}^{K} w_k (1 - \tilde{\lambda})^k$.

**BernNet.** He et al. [21] exploit Bernstein basis $\binom{K}{k}(1-x)^{K-k}x^k$ as the polynomial of the $k$-th layer. By piling up $K$ such convolution layers with trainable parameter $w_k$, they propose BernNet as

$$\mathbf{z} = \sum_{k=0}^{K} \frac{w_k}{2^K}\binom{K}{k}(2\mathbf{I} - \mathcal{L})^{K-k}\mathcal{L}^k \cdot f_\theta(\mathbf{x}), \quad (4)$$

Intuitively, the filter function of BernNet is $\mathrm{g}_{\mathbf{w}}(\lambda) = \sum_{k=0}^{K} \frac{w_k}{2^K}\binom{K}{k}(2-\lambda)^{K-k}\lambda^k$.

**JacobiConv.** Recently, Wang and Zhang [44] leverage Jacobi polynomial [1] bases $P_k^{a,b}(x)$, a general orthogonal polynomials. They devise JacobiConv using Jacobi polynomials as

$$\mathbf{z} = \sum_{k=0}^{K} w_k P_k^{a,b}(\mathbf{I} - \mathcal{L}) \cdot \mathbf{x}, \quad (5)$$

where $w_k$ is the trainable parameter. The corresponding filter function is $\mathrm{g}_{\mathbf{w}}(\lambda) = \sum_{k=0}^{K} w_k P_k^{a,b}(1 - \lambda)$.

## 3 POLYNOMIAL FILTERS REVISIT

### 3.1 Polynomial Filters in Krylov Subspace

The four polynomials introduced in Section 2.2 are essentially four different linearly independent bases of polynomial vector spaces. By simply reordering the terms of the four filter functions, they can be reformulated equally by the monomial sequence $\{1, x, x^2, \cdots, x^K, \cdots\}$, i.e.,

$$\mathbf{z} = \sum_{k=0}^{K} \mathbf{w}_k \mathbf{P}^k \cdot \mathbf{x} \quad (6)$$

where $K$ is the predefined degree of the polynomial with $K \ll n$, $\mathbf{w}_k$ is the $k$-th learnable parameter with $\mathbf{w} \in \mathbb{R}^{K+1}$, and $\mathbf{P}$ is the corresponding propagation matrix. From the perspective of polynomial vector space, the learned polynomial filter $\sum_{k=0}^{K} \mathbf{w}_k \mathbf{P}^k$ is the combination of $K+1$ polynomial bases of the space with coefficients $\mathbf{w}_k$ for $k \in \{0, \cdots, K\}$. However, if regarding the product $\mathbf{P}^k \mathbf{x}$ as the $k$-th vector basis, $\mathbf{P}^k \mathbf{x}$ for $k \in \{0, 1, \cdots, K\}$ forms the order-$(K+1)$ *Krylov Subspace* $\mathcal{K}_{K+1}(\mathbf{P}, \mathbf{x}) = \mathrm{span}\{\mathbf{x}, \mathbf{P}\mathbf{x}, \mathbf{P}^2\mathbf{x}, \cdots, \mathbf{P}^K\mathbf{x}\}$, and the filtered signal $\mathbf{z} = \sum_{k=0}^{K} \mathbf{w}_k \mathbf{P}^k \mathbf{x}$ is deemed as the weighted combination of the bases of $\mathcal{K}_{K+1}(\mathbf{P}, \mathbf{x})$. As such, we establish the following proposition.

PROPOSITION 1. *Given a graph* $\mathbf{G}$ *and a graph signal* $\mathbf{x}$, *the set of all $K$-order polynomial filters with propagation matrix* $\mathbf{P}$ *belongs to the Krylov subspace* $\mathcal{K}_{K+1}(\mathbf{P}, \mathbf{x})$.

The aim of polynomial filters is to approximate the spectral filter $\mathbf{U}\mathrm{g}_{\mathbf{w}}(\mathbf{\Lambda})\mathbf{U}^\top \mathbf{x}$ so as to avoid eigendecomposition. In what follows, we prove that the spectral filter can be *equivalently* expressed by polynomial filters constructed from a Krylov subspace with order $K$ determined by both matrix $\mathbf{P}$ and vector $\mathbf{x}$. Before proceeding further, we first introduce the concept of vector grade with respect to a matrix.

DEFINITION 2 (GRADE OF $\mathbf{x}$ WITH RESPECT TO $\mathbf{P}$ [29]). *The grade of* $\mathbf{x}$ *with respect to* $\mathbf{P}$ *is a positive integer $t$ such that*

$$\dim \mathcal{K}_K(\mathbf{P}, \mathbf{x}) = \min(K, t).$$

The grade $t$ of $\mathbf{x}$ with respect to $\mathbf{P}$ defines the highest dimension of a Krylov subspace constructed by $\mathbf{P}$ and $\mathbf{x}$. Ergo, the new vectors $\mathbf{P}^K \mathbf{x}$ are *linearly dependent* on the previous ones and $\mathcal{K}_K(\mathbf{P}, \mathbf{x}) = \mathcal{K}_{K+1}(\mathbf{P}, \mathbf{x})$ for $K \geq t$. As a consequence, we have the following conclusion relating to the optimal spectral filter.

PROPOSITION 2. *Consider a propagation matrix* $\mathbf{P} = \mathbf{I} - \mathcal{L}$ *and a graph signal* $\mathbf{x}$ *with grade $t$ with respect to* $\mathbf{P}$. *There exists a* $(t-1)$-*order polynomial filter* $\sum_{k=0}^{t-1} \theta_k \mathbf{P}^k \mathbf{x}$ *parameterized by* $\theta \in \mathbb{R}^t$ *equivalent to the optimal spectral filter* $\mathbf{U}\mathrm{g}_{\mathbf{w}}(\mathbf{\Lambda})\mathbf{U}^\top \mathbf{x}$.

Proposition 2 implies that the utilization of $K$-order polynomial filters approximates the spectral filter by preserving only the first $K$ bases from the order-$t$ Krylov subspace $\mathcal{K}_t(\mathbf{P}, \mathbf{x})$ and discarding the rest bases.

## 3.2 Theoretical Analysis on Bases Convergence

As introduced in Section 2.2, polynomial filters utilize different propagation matrices $\mathbf{P}$. To elaborate, ChebNet sets $\mathbf{P} = 2\mathcal{L}/\lambda_{max} - \mathbf{I}$ where $\lambda_{max}$ is the largest eigenvalue of $\mathcal{L}$. GPR-GNN exploits $\mathbf{P} = \mathbf{I} - \tilde{\mathcal{L}}$ where $\tilde{\mathcal{L}} = \mathbf{I} - \tilde{\mathbf{D}}^{-\frac{1}{2}} \tilde{\mathbf{A}} \tilde{\mathbf{D}}^{-\frac{1}{2}}$. BernNet employs $\mathbf{P} = \mathbf{I} - \mathcal{L}/2$ while JacobiConv adopts $\mathbf{P} = \mathbf{I} - \mathcal{L}$. In general, propagation matrix $\mathbf{P}$ is derived from the Laplacian matrix $\mathcal{L}$ or its variant $\tilde{\mathcal{L}}$. Typically, eigenvalues of $\mathbf{P}$ lie in the interval $(-1, 1]^2$.

Towards polynomial bases, one pertinent and crucial problem is the convergence determined by propagation matrix $\mathbf{P}$, which in turn influences the selections of polynomial order $K$. To this end, we first prove that for a sufficiently large $K$, the basis vector $\mathbf{P}^K \mathbf{x}$ converges to $\phi \phi^\top \mathbf{x}$ where $\phi$ is the eigenvector associated with the largest eigenvalue of $\mathbf{P}$. Consequently, we establish an upper bound for the choice of an appropriate $K$ in terms of $\mathbf{P}$ measured by the *relative pointwise distance* [39].

THEOREM 1. *Let $\lambda_i(\mathbf{P})$ be the $i$-th eigenvalue of the matrix $\mathbf{P}$ where $i \in \{1, 2, \cdots, n\}$ and $\lambda^* := \max\{-\lambda_1(\mathbf{P}), \lambda_{n-1}(\mathbf{P})\}$. It holds that (i) $\mathbf{P}^K$ converges asymptotically to a stable propagation matrix $\mathbf{P}_\pi$, i.e., $\mathbf{P}_\pi := \lim_{K \to \infty} \mathbf{P}^K$; and (ii) given $\epsilon \in (0, 1)$, $\max_{u,v \in \mathcal{V}} |\mathbf{P}^K[u, v] - \mathbf{P}_\pi[u, v]| \leq \epsilon \mathbf{P}_\pi[u, v]$ if $K \geq \ln \frac{\epsilon d_{\min}}{2m} / \ln \lambda^*$ where $d_{\min} := \min_{v \in V} d_v$.*

Theorem 1 elucidates that the convergence speed of the polynomial basis is dominated by value $\lambda^*$, i.e., the second largest *absolute* value among eigenvalues of the propagation matrix $\mathbf{P}$. In particular, a smaller $\lambda^*$ corresponds to a more rapid convergence speed. Moreover, the empirical efficacy of polynomial filters is substantially determined by $\mathbf{P}$ which plays a pivotal role in the construction of Krylov subspace. Yet, propagation matrices in existing polynomials asymptotically converge towards stability regardless of the heterophily degrees of underlying graphs. As a consequence, a crucial question is *how we can optimize the design of the propagation matrix to enhance the expressive power of polynomial filters*. We will answer this question in Section 4.

## 3.3 Connection to Spatial GNNs

Polynomial graph filters approximate the underlying spectral graph filters by leveraging different polynomials. By identifying the propagation matrices $\mathbf{P}$, they can be expressed equivalently as $\sum_{k=0}^{K-1} w_k \mathbf{P}^k \mathbf{x}$. The weight parameter $w_k$ inherently stems from the coefficients of original polynomials and learnable weights. From a spatial perspective, the order $K$ of propagation matrix $\mathbf{P}$ is the propagation step in feature aggregation from neighbors. Therefore, the fundamental distinction among diverse polynomials lies in that they assign varying weights to neighbors across multiple hops. Meanwhile, as analyzed in Section 3.2, different propagation matrices $\mathbf{P}$ exhibit divergent convergence properties. Those disparities lead to distinct empirical performance, attributed to the different capabilities of the polynomials to capture the intricate characteristics of underlying graphs.

Proposition 2 reveals that propagation within the grade $t$ hops of signal $\mathbf{x}$ with respect to propagation matrix $\mathbf{P}$ suffices. This fact hints that $\forall v \in \mathcal{V}$ receives all signals from neighbors within $\mathbf{t}$ hops

---

$^2$$-1$ is achievable when graphs are bipartite, which is not considered in our paper.

and information beyond $t$ hops contributes no information gain. For example, let $\mathbf{x}$ be one eigenvector of $\mathbf{P}$ and we get the grade $t = 1$. Therefore, signals received out of the first step are the same as those received from the one-hop neighbors.

# 4 POLYNOMIAL FILTER FROM ADAPTIVE KRYLOV SUBSPACE

## 4.1 Adaptive Krylov Subspace

As proven, existing polynomial filters belong to the Krylov subspace $\mathcal{K}(\mathbf{P}, \mathbf{x})$ and have theoretically equivalent expressive power, wherein the propagation matrix $\mathbf{P}$ plays a crucial role. Despite that, matrix $\mathbf{P}$ in existing filters is predefined and fixed regardless of the homophily ratios of underlying graphs. Yet, responded frequencies are significantly correlated with homophily ratios.

For ease of exposition, we consider a binary node classification for a graph $\mathbf{G} = (\mathcal{V}, \mathcal{E})$ in homophily ratio $h$ associated with a node signal $\mathbf{x} \in \mathbb{R}^n$. Let $\mathbf{y}_0 \in \mathbb{R}^n$ (resp. $\mathbf{y}_1 \in \mathbb{R}^n$) be the one-hot label vector such that $\mathbf{y}_{0,i} = 1$ (resp. $\mathbf{y}_{1,i} = 1$) if node $u_i$ belongs to class 0 (resp. class 1), otherwise $\mathbf{y}_{0,i} = 0$ (resp. $\mathbf{y}_{1,i} = 0$). Let $\mathbf{y} = \mathbf{y}_0 - \mathbf{y}_1$ be the label differences. Therefore, we can establish that

$$\frac{\mathbf{y}^\top \mathcal{L} \mathbf{y}}{\mathbf{y}^\top \mathbf{y}} = \frac{(\mathbf{D}^{1/2}\mathbf{y})^\top (\mathbf{I} - \mathbf{D}^{-1/2}\mathbf{A}\mathbf{D}^{-1/2})(\mathbf{D}^{1/2}\mathbf{y})}{(\mathbf{D}^{1/2}\mathbf{y})^\top (\mathbf{D}^{1/2}\mathbf{y})} = \frac{\mathbf{y}^\top \mathbf{D}^{1/2}(\mathbf{I} - \mathbf{D}^{-1/2}\mathbf{A}\mathbf{D}^{-1/2})\mathbf{D}^{1/2}\mathbf{y}}{\sum_{i=1}^n d_i \mathbf{y}_i^2}$$

$$= \frac{\mathbf{y}^\top (\mathbf{D} - \mathbf{A})\mathbf{y}}{\sum_{i=1}^n d_i \mathbf{y}_i^2} = \frac{\sum_{\langle u_i, u_j \rangle \in \mathcal{E}} (\mathbf{y}_i - \mathbf{y}_j)^2}{\sum_{i=1}^n d_i \mathbf{y}_i^2} = \frac{4(1-h)m}{m} = 4(1-h).$$

Since $\mathbf{y}^\top \mathbf{y} = n$, we have $4(1-h)n = \mathbf{y}^\top \mathcal{L} \mathbf{y}$. Meanwhile, let $\mathcal{L} = \mathbf{U} \Lambda \mathbf{U}^\top$ where $\mathbf{U}$ is the eigenvector matrix and $\Lambda$ is the diagonal matrix of eigenvalue spectrum $\{\lambda_1, \lambda_2, \cdots, \lambda_n\}$. W.l.o.g., suppose $\mathbf{U}^\top \mathbf{y} = (\alpha_1, \alpha_2, \cdots, \alpha_n)$ where $\alpha_i$ is the projection (response) of $\mathbf{y}$ on eigenvector $\mathbf{U}^\top[i, :]$. Then we have $\mathbf{y}^\top \mathcal{L} \mathbf{y} = \mathbf{y}^\top \mathbf{U} \Lambda \mathbf{U}^\top \mathbf{y} = \sum_{i=1}^n \alpha_i^2 \lambda_i$. Therefore, we have $\frac{1}{n} \sum_{i=1}^n \alpha_i^2 \lambda_i = 4(1-h)$, i.e., graphs with different homophily ratios respond to different frequencies.

This fact motivates us to design an adaptive $\mathbf{P}$ that provides better adaptability and controllability. Recall that the propagation matrix $\mathbf{P}$ is primarily derived from the Laplacian matrix $\mathcal{L}$ and explored in the graph heat diffusion. Hence, we resort to the following *Graph Heat Equation* [9, 45].

$$\frac{d\mathbf{H}_t}{dt} = -\mathcal{L}\mathbf{H}_t, \qquad \mathbf{H}_0 = \mathbf{X}, \tag{7}$$

where $\mathbf{H}_t$ is the node representations of graph $\mathbf{G}$ at time $t$. Given an infinitesimal interval $\tau$, the Euler method implies that

$$\mathbf{H}_{t+\tau} = \lim_{\tau \to 0^+} \mathbf{H}_t - \tau \mathcal{L} \mathbf{H}_t$$

$$= \lim_{\tau \to 0^+} \mathbf{H}_t (\mathbf{I} - \tau \mathcal{L})$$

$$= \lim_{\tau \to 0^+} \mathbf{H}_t ((1-\tau)\mathbf{I} + \tau \mathbf{D}^{-\frac{1}{2}} \mathbf{A} \mathbf{D}^{-\frac{1}{2}}). \tag{8}$$

Equation (8) demonstrates that $(1-\tau)\mathbf{I} + \tau \mathbf{D}^{-\frac{1}{2}}\mathbf{A}\mathbf{D}^{-\frac{1}{2}}$ propagates $\mathbf{H}_t$ forward with a stride $\tau$. In light of this, we design our propagation matrix $\mathbf{P}_\tau$ via *renormalization trick* [23] as

$$\mathbf{P}_\tau = (\tau\mathbf{D} + (1-\tau)\mathbf{I})^{-\frac{1}{2}}(\tau\mathbf{A} + (1-\tau)\mathbf{I})(\tau\mathbf{D} + (1-\tau)\mathbf{I})^{-\frac{1}{2}} = \mathbf{D}_\tau^{-\frac{1}{2}} \mathbf{A}_\tau \mathbf{D}_\tau^{-\frac{1}{2}},$$

where $\mathbf{A}_\tau = \tau\mathbf{A} + (1-\tau)\mathbf{I}$ and $\mathbf{D}_\tau = \tau\mathbf{D} + (1-\tau)\mathbf{I}$. In particular, $\tau$ decides the percentages of signals from ego parts and neighbors in feature aggregations. In the meantime, the $\tau$ value is able to alter the graph spectrum as follows.

THEOREM 2. *Let $\mathcal{L}_\tau = \mathbf{I} - \mathbf{P}_\tau$ and $\lambda_i(\tau)_{i=1}^n$ denote the eigenvalues of $\mathcal{L}_\tau$ with $\lambda_1(\tau) \leq \lambda_2(\tau) \leq \cdots \leq \lambda_n(\tau)$ for $\tau > 0$. Then it holds that $\lambda_i(\tau)$ is a monotonically increasing function of $\tau$ and $\lambda_i(\tau) \leq \lambda_i$ if $\tau \in (0, 1]$ and $\lambda_i(\tau) > \lambda_i$ if $\tau > 1$ for $i \in [n]$.*

Theorem 2 reveals that changing $\tau$ is capable of reshaping the underlying spectrum of $\mathcal{L}_\tau$. In doing so, we are able to build *adaptive Krylov subspace* $\mathcal{K}_{K+1}(\mathbf{P}_\tau, \mathbf{x})$ such that the corresponding Krylov basis, known as *adaptive Krylov basis*, has the adaptability to graphs with varying homophily ratios.

## 4.2 Polynomial Filter AdaptKry

In this section, we propose AdaptKry, an optimized polynomial filter utilizing the adaptive Krylov basis from the adaptive Krylov subspace $\mathcal{K}_K(\mathbf{P}_\tau, \mathbf{x})$. Specifically, AdaptKry first initializes a weight vector $\mathbf{w} \in \mathbb{R}^{K+1}$ and then constructs the Krylov basis $\{\mathbf{x}, \mathbf{P}_\tau \mathbf{x}, \cdots, \mathbf{P}_\tau^K \mathbf{x}\}$. Next, those vectors are concatenated and multiplied by $\mathbf{w}$ in an element-wise manner, after which it is fed into an arbitrary Multilayer Perceptron (MLP) network to train the weight parameter $\mathbf{w}$. The pseudo-code for AdaptKry is illustrated in Algorithm 1. It is worth mentioning that AdaptKry, in contrast to existing polynomial filters, separates the procedures of feature propagation and weight learning. This decoupling mechanism enables adaptive Krylov bases unchanged during training and thus averts the repetitive calculation on polynomials.

**AdaptKry Extension.** Complex networks often consist of numerous components characterized by diverse spectral properties. In such scenarios, a singular adaptive Krylov basis might lack the expressiveness necessary to capture the varying spectral characteristics. To address this challenge, it is prudent to utilize multiple bases derived from distinct adaptive Krylov spaces. For instance, one can employ appropriate $r \in \mathbb{N}$ bases with different $\tau$ values as

$$
\begin{aligned}
\mathbf{z} &= \sum_{k=0}^K \mathbf{w}_k^1 \mathbf{P}_{\tau_1}^k \cdot \mathbf{x} + \cdots + \sum_{k=0}^K \mathbf{w}_k^r \mathbf{P}_{\tau_r}^k \cdot \mathbf{x} \\
&= \sum_{k=0}^K \left( \sum_{i=1}^r \frac{\mathbf{w}_k^i}{r} \mathbf{P}_{\tau_i}^k \right) \cdot \mathbf{x} \\
&= \sum_{k=0}^K \mathbf{w}_k \left( \sum_{i=1}^r \mathbf{P}_{\tau_i}^k \right) \cdot \mathbf{x}.
\end{aligned}
\tag{9}
$$

By integrating the weight parameters $\sum_{i=1}^r \mathbf{w}_k^i / r$ into a singular trainable parameter $\mathbf{w}_k$, it enhances the expressive capability without escalating the training burden. Meanwhile, the specific number and values of the set of $\tau$ parameter $\{\tau_1, \tau_2, \cdots, \tau_r\}$ facilitate an enriching analysis the complexity of networks with heterogeneous homophily ratios.

**Space & Time Complexity.** In AdaptKry, the space consumption to store $\mathbf{A}_\tau$, $\mathbf{D}_\tau$, and $\mathbf{P}_\tau$ is $2m + 3n$ in total. The space costs of feature matrix $\mathbf{X}$ and Krylov basis are $nd$ and $Knd$ respectively. Thus the space complexity of AdaptKry is $O(m + Knd)$. The running time of AdaptKry is dominated by the computation of Krylov basis, i.e., $K$ times of $\mathbf{P}_\tau \cdot \mathbf{F}^{(\ell-1)}$ (Line 5 in Algorithm 1), which is sparse matrix multiplication at the cost of $(m+n)d$. Thus the total time complexity of AdaptKry is $O(K(m+n)d)$.

---

**Algorithm 1:** AdaptKry

**Input:** Graph $\mathbf{G}$, feature matrix $\mathbf{X}$, order $K$, step size $\tau$
**Output:** $\mathbf{Z}$

1  $\mathbf{Z} \leftarrow \mathbf{X}$, $\mathbf{F}^{(0)} \leftarrow \mathbf{X}$;
2  Weight vector $\mathbf{w}$ initialization;
3  $\mathbf{A}_\tau \leftarrow \tau \mathbf{A} + (1-\tau)\mathbf{I}$, $\mathbf{D}_\tau \leftarrow \tau \mathbf{D} + (1-\tau)\mathbf{I}$, $\mathbf{P}_\tau \leftarrow \mathbf{D}_\tau^{-\frac{1}{2}} \mathbf{A}_\tau \mathbf{D}_\tau^{-\frac{1}{2}}$;
4  **for** $\ell \leftarrow 1$ **to** $K$ **do**
5     $\mathbf{F}^{(\ell)} \leftarrow \mathbf{P}_\tau \cdot \mathbf{F}^{(\ell-1)}$;
6     $\mathbf{Z} \leftarrow \text{concate}(\mathbf{Z}, \mathbf{F}^{(\ell)})$;
7  $\mathbf{Z} \leftarrow \text{MLP}(\mathbf{Z} \odot \mathbf{w})$;
8  **return** $\mathbf{Z}$;

---

## 4.3 Approximation Analysis

Theoretically, AdaptKry provides equivalent expressive power with existing polynomial graph filters. However, by equipping adaptability to AdaptKry, it provides the ability to alter the underlying spectrum as desired. In this regard, AdaptKry derived from adaptive Krylov subspace yields an improved capability to capture the spectral characteristics of underlying graphs, compared to existing polynomial filters from ordinary Krylov subspace. Furthermore, we analyze in Section 3.2 that the convergence rate of polynomials is governed by value $\lambda^*$ determined by eigenvalues of propagation matrices. Therefore, it is intuitive that reshaping the underlying spectrum enhances the expressive capabilities of polynomials.

THEOREM 3. *Consider a connected graph $\mathbf{G} = (\mathcal{V}, \mathcal{E})$ with graph signal $\mathbf{x} \in \mathbb{R}^n$. Let $\mathbf{F}_1(\mathbf{x}) = \sum_{k=0}^{t-1} \mathbf{w}_k^o \mathbf{P}^k \mathbf{x}$ and $\mathbf{F}_2(\mathbf{x}) = \sum_{k=0}^{t-1} \mathbf{w}_k^a \mathbf{P}_\tau^k \mathbf{x}$ are the optimal polynomial filters from ordinary Krylov subspace $\mathcal{K}_t(\mathbf{P}, \mathbf{x})$ and adaptive Krylov subspace $\mathcal{K}_t(\mathbf{P}_\tau, \mathbf{x})$ respectively where $t$ is the grade of graph signal to the propagation matrix[3]. When training models from the two Krylov subspaces, we assume the initial weight $\mathbf{w}$ is randomly sampled from a predefined distribution. It holds that $\mathbb{E}[\|\mathbf{w}^a - \mathbf{w}\|_2] \leq \mathbb{E}[\|\mathbf{w}^o - \mathbf{w}\|_2]$ where the expectation is over the randomness of $\mathbf{w}$.*

**Discussion on basis orthogonality.** Theoretically, distinct polynomial bases yield equivalent expressive capabilities, as demonstrated in Proposition 1. The major distinction lies in the convergence rates of gradient descent in the process of model training. The convergence rate reaches optimal if the condition number of the Hessian matrix of gradients is minimal, as demonstrated in [4, 46]. As for polynomial filters, the condition number becomes minimum when polynomials form an orthonormal basis with respect to the underlying weight function [44][4] which, however, is intractable. Instead, as stated in [17, 29], such orthonormal basis can be directly established by orthogonalizing the basis of $\mathcal{K}_K(\mathbf{P}, \mathbf{x})$ via *three-term recurrence* method [15, 29] without resolving the weight function. Despite that, there are two potential weaknesses. First, similar to existing polynomials, orthonormal bases demonstrate constrained flexibility in accommodation to graphs with diverse heterophily, thus yielding suboptimal empirical performance, as confirmed in our ablation study in Section 5.3. Second, the orthogonalization of the basis incurs extra $O(Kn)$ computation costs.

---

[3]According to Definition 2, it is intuitive that the two subspaces share the same grade $t$ irrespective of $\tau$.

[4]Notice that we cannot claim the polynomial basis is orthonormal without specifying the weight function.

Table 2: Dataset Details.

| Dataset | Cora | Citeseer | Pubmed | Actor | Chameleon | Squirrel |
|---|---|---|---|---|---|---|
| #Nodes ($n$) | 2,708 | 3,327 | 19,717 | 7,600 | 2,277 | 5,201 |
| #Edges ($m$) | 5,429 | 4,732 | 44,338 | 26,659 | 31,371 | 198,353 |
| #Features ($d$) | 1,433 | 3,703 | 500 | 932 | 2,325 | 2,089 |
| #Classes | 7 | 6 | 3 | 5 | 5 | 5 |
| Homo. ratio ($h$) | 0.81 | 0.73 | 0.80 | 0.22 | 0.23 | 0.22 |

## 5 EXPERIMENTS

In this section, we aim to i) evaluate the capability of AdaptKry as a graph filter for node classification on both homophily and heterophily graphs against a series of baselines and ii) validate the design of AdaptKry and the properties of adaptive Krylov bases. The classification performance is measured in accuracy.

### 5.1 Experimental Setting

**Datasets.** We evaluate the methods on 6 real-world datasets with varied sizes and homophily ratios, as presented in Table 2. In particular, the 3 citation networks [40], i.e., Cora, Citeseer, and Pubmed, are homophily graphs with homophily ratios 0.81, 0.73, and 0.80 respectively; the Wikipedia graphs Chameleon and Squirrel, the Actor co-occurrence graph from WebKB3 [34] are heterophily graphs with homophily ratios 0.22, 0.23, and 0.22 respectively.

By following the convention [20, 21, 44], we i) generate 10 random splits of training/validation/testing with 60%/20%/20% percents respectively for the 6 datasets and report the average accuracy with associated standard deviations for each method.

**Algorithms.** We consider two seminal GNN models GCN [23] and the simplified SGC [47], and also include SIGN [14] and ASGC [6] as they are two variants of SGC for simplicity. Five polynomial graph filters, i.e., GPR-GNN [8], EvenNet [26], ChebNet [10] and its improved version ChebNetII [20], BernNet [21], and Jacobi-Conv [44], and one recently proposed orthogonal basis model Opt-BasisGNN [17] are deemed as the competitors of polynomial filters. We compare our AdaptKry against 11 baselines on the 6 datasets. Appendix A.2 provides the links to the source codes of baselines.

**Parameter Settings.** There are 4 key hyperparameters in AdaptKry, i.e., the neighbor hop $K$, the step size $\tau$, learning rate and hidden dimension. For a fair comparison, we fix $K = 10$ for all tested methods. We tune these parameters to acquire the best attainable performance on each graph according to their homophily ratios, as summarized in Table 6 in Appendix A.2. For baselines, we either adopt the public results if applicable or follow the original settings in [21, 44, 47] for the best possible performance. More details are discussed in Appendix A.2.

### 5.2 Node Classification

Table 3 presents the accuracy associated with the standard deviation of all tested 12 methods on the 6 datasets with homophily ratios from $0.22 \sim 0.81$. For ease of exposition, we highlight the *highest* accuracy score in bold and underline the *second highest* accuracy score for each dataset.

First of all, observe that AdaptKry achieves the highest accuracy scores almost on all datasets except a slight lag on Squirrel. In particular, AdaptKry advances the accuracy score up to 3.0% and 1.18%

on the homophily datasets Citeseer and Pubmed. These observations strongly support the state-of-the-art capability of AdaptKry as polynomial filters by utilizing the adaptive Krylov basis. Meanwhile, it also confirms our analysis in Section 4.3. As analyzed in Section 4.1, the notable advantage of AdaptKry arises from the adaptive parameter $\tau$. This enables AdaptKry to adapt graphs with a wide range of homophily ratios as various heterophily graphs suit different $\tau$ values, further explored in Section 5.3.

### 5.3 Ablation Study

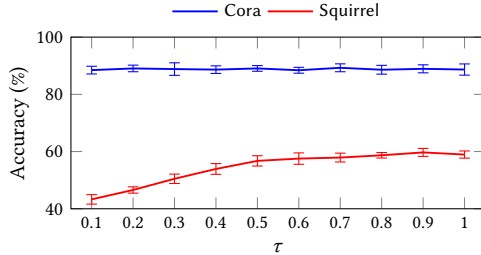

Figure 2: Accuracy of AdaptKry with varying $\tau$ values.

**Sensitivity on parameter $\tau$.** As stated in Section 4.1, $\tau$ in propagation matrix $\mathbf{P}_\tau$ controls the portions of signals from ego parts and neighbors in the feature aggregation. To evaluate the sensitivity of AdaptKry towards $\tau$ for different datasets, we test AdaptKry by setting $\tau = \{0.1, 0.2, 0.3, \cdots, 0.9, 1.0\}$ on one homophily graph Cora (homo. ratio 0.81) and one heterophily graph Squirrel (homo. ratio 0.22). The resultant accuracy scores are plotted in Figure 2. As shown, the accuracy scores on Cora remain close for varying $\tau$, whilst those on Squirrel rise along with the increase of $\tau$ and reach the peak at $\tau = 0.9$. This reveals that heterophily graphs are more sensitive to the $\tau$ value compared with homophily graphs since feature signals of connected neighbors from heterophily graphs differ more. This result further confirms our claim that graphs with different homophily ratios respond to different frequencies in Section 4.1.

**Spectrum reshaping by $\tau$.** As proved in Theorem 2, the parameter $\tau$ within the propagation matrix $\mathbf{P}_\tau$ is able to reshape the graph spectrum, thereby offering underlying polynomial basis the adaptability. To elaborate the influence of $\tau$ on polynomial bases, we generate three sets of order-$K$ polynomial bases with $\tau = \{0.5, 1.0, 1.5\}$ for Cora and Squirrel. The resultant bases $\mathbf{Z}$ is $\mathbf{Z} \in \mathbb{R}^{n \times (K+1)d}$ for each $\tau$ value where $d$ is the feature dimension. Specifically, for feature signal $\mathbf{x} \in \mathbb{R}^n$ from each dimension of feature matrix $\mathbf{X} \in \mathbb{R}^{n \times d}$, we compute the angles between signal vectors in two consecutive hops within the basis, leading to a total of $Kd$ angles. Subsequently, these angles are averaged for the $d$ signals, resulting in $K$ angles that characterize the polynomial basis. In particular, we keep $K = 10$ for this experiment.

Figure 3 illustrates the evolving patterns of angles between two consecutive signal vectors in polynomial bases across three different $\tau$ values. Observe that polynomial bases with $\tau = 0.5$ and $\tau = 1.0$ all converge steadily, which is consistent with Theorem 1. In particular, basis with $\tau = 0.5$ converges faster than that

**Table 3: Accuracy (%) on small datasets.**

| Methods | Cora | Citeseer | Pubmed | Actor | Chameleon | Squirrel |
|---|---|---|---|---|---|---|
| GCN | 87.18 ± 1.12 | 79.85 ± 0.78 | 86.79 ± 0.31 | 33.26 ± 1.15 | 60.81 ± 2.95 | 45.87 ± 0.88 |
| SGC | 86.83 ± 1.28 | 79.65 ± 1.02 | 87.14 ± 0.90 | 34.46 ± 0.67 | 44.81 ± 1.20 | 25.75 ± 1.07 |
| ASGC | 85.35 ± 0.98 | 76.52 ± 0.36 | 84.17 ± 0.24 | 33.41 ± 0.80 | 71.38 ± 1.06 | 57.91 ± 0.89 |
| SIGN | 87.70 ± 0.69 | 80.14 ± 0.87 | 89.09 ± 0.43 | 41.22 ± 0.96 | 60.92 ± 1.45 | 45.59 ± 1.40 |
| ChebNet | 87.32 ± 0.92 | 79.33 ± 0.57 | 87.82 ± 0.24 | 37.42 ± 0.58 | 59.51 ± 1.25 | 40.81 ± 0.42 |
| GPR-GNN | 88.54 ± 0.67 | 80.13 ± 0.84 | 88.46 ± 0.31 | 39.91 ± 0.62 | 67.49 ± 1.38 | 50.43 ± 1.89 |
| BernNet | 88.51 ± 0.92 | 80.08 ± 0.75 | 88.51 ± 0.39 | 41.71 ± 1.12 | 68.53 ± 1.68 | 51.39 ± 0.92 |
| JacobiConv | 88.98 ± 0.72 | 80.78 ± 0.79 | 89.62 ± 0.41 | 41.17 ± 0.64 | 74.20 ± 1.03 | 57.38 ± 1.25 |
| EvenNet | 87.77 ± 0.67 | 78.51 ± 0.63 | 90.87 ± 0.34 | 40.36 ± 0.65 | 67.02 ± 1.77 | 52.71 ± 0.85 |
| ChebNetII | 88.71 ± 0.93 | 80.53 ± 0.79 | 88.93 ± 0.29 | 41.75 ± 1.07 | 71.37 ± 1.01 | 57.72 ± 0.59 |
| OptBasisGNN | 87.00 ± 1.55 | 80.58 ± 0.82 | 90.30 ± 0.19 | 42.39 ± 0.52 | 74.26 ± 0.74 | **63.62 ± 0.76** |
| AdaptKry | **89.95 ± 0.95** | **83.78 ± 0.38** | **92.05 ± 0.25** | **42.70 ± 1.14** | **74.53 ± 1.21** | 63.31 ± 0.76 |

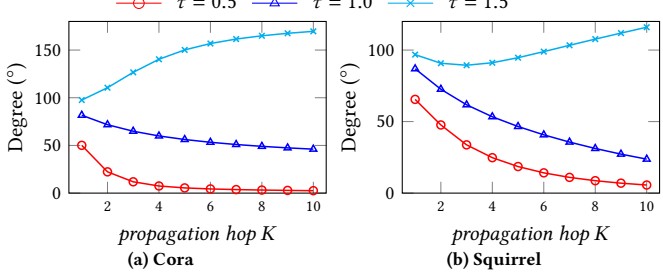

**Figure 3: Bases reshaping by $\tau$.**

**Table 4: Accuracy (%) and corresponding $\tau$ set.**

| $\tau$ set | Cora | Actor | Chameleon | Squirrel |
|---|---|---|---|---|
| $\{\tau_1\}$ | 88.90 ± 1.70 | 42.06 ± 1.27 | 74.09 ± 1.22 | **63.31 ± 0.76** |
| $\{\tau_1, \tau_2\}$ | 89.85 ± 1.36 | 42.16 ± 1.73 | **74.53 ± 1.21** | 63.31 ± 0.76 |
| $\{\tau_1, \tau_2, \tau_3\}$ | **89.95 ± 0.95** | **42.70 ± 1.14** | 73.83 ± 0.77 | 63.31 ± 0.76 |
| $\{\tau_1\}$ | {0.8} | {0.7} | {0.8} | {0.8} |
| $\{\tau_1, \tau_2\}$ | {0.8, 1.0} | {0.6, 1.4} | **{0.5, 0.8}** | {0.8, 0.8} |
| $\{\tau_1, \tau_2, \tau_3\}$ | **{0.5, 0.8, 1.1}** | **{0.6, 1.7, 1.8}** | {0.5, 0.8, 1.1} | {0.8, 0.8, 0.8} |

**Table 5: Accuracy (%) of orthogonal / non-orthogonal polynomials**

| Dataset | Method | hop $K$ | | | | |
|---|---|---|---|---|---|---|
| | | 2 | 4 | 6 | 8 | 10 |
| Cora | OrthKry | 87.37 | 88.41 | 88.60 | 87.91 | 86.09 |
| | AdaptKry | 87.85 | 89.64 | 88.75 | 89.67 | 88.90 |
| Citeseer | OrthKry | 81.88 | 81.46 | 81.43 | 80.90 | 80.24 |
| | AdaptKry | 83.82 | 83.32 | 83.44 | 83.66 | 83.78 |
| Chameleon | OrthKry | 72.14 | 70.42 | 74.90 | 72.99 | 72.70 |
| | AdaptKry | 68.84 | 72.56 | 72.39 | 72.54 | 73.22 |
| Squirrel | OrthKry | 62.21 | 62.61 | 63.13 | 64.65 | 65.14 |
| | AdaptKry | 60.31 | 61.90 | 62.23 | 63.15 | 63.31 |

of $\tau = 1.0$. This is due to the fact that $\lambda^*(0.5) < \lambda^*(1.0)$ where $\lambda^*(\tau)$ is the second largest *absolute* value among eigenvalues of $\mathbf{P}_\tau$ where $\mathbf{P}_\tau = \mathbf{I} - \mathcal{L}_\tau$. According to Theorem 1, a smaller $\lambda^*$ indicates faster convergence. To explain, Theorem 2 proves that $\lambda_i(0.5) \le \lambda_i(1.0)$ where $\lambda_i(\tau)$ is the $i$-th eigenvalue of Laplacian matrix $\mathcal{L}_\tau$. In real-world datasets with $\tau = 1.0$, $\lambda^*(1.0) = |1-\lambda_{n-1}(1.0)|$ since $|1 - \lambda_{n-1}(1.0)| \ge |1 - \lambda_2(1.0)|$. However, when $\tau = 0.5$, $\lambda_{n-1}(0.5)$ becomes substantially smaller than $\lambda_{n-1}(1.0)$ such that $|1-\lambda_{n-1}(0.5)| \le |1-\lambda_2(0.5)| = \lambda^*(0.5) < |1-\lambda_{n-1}(1.0)| = \lambda^*(1.0)$. Note that polynomial bases with $\tau = 1.5$ diverge as the propagation hop $K$ increases. This divergence occurs due to $\lambda^*(1.5) > 1$, leading to a significant divergence in the polynomial basis $\mathbf{P}^K \mathbf{x}$.

**Extension of AdaptKry.** To demonstrate the enhanced expressive power of extended AdaptKry, we adopt polynomial bases with the number in the set of $\{1, 2, 3\}$ and test on datasets Cora, Actor, Chameleon, and Squirrel. To obtain the best $\tau$ set on each dataset, we tune the set of $\tau$ values in the set of $\{0.1, 0.2, \cdots, 1.8, 1.9\}$.

Table 4 presents the obtained accuracy scores and the corresponding $\tau$ set. We highlight the highest score and the associated $\tau$ set in bold. When achieving the highest scores, AdaptKry adopts *three* polynomial bases on both Cora and Actor and uses *two* and *one* bases on Chameleon and Squirrel respectively. The exact number and value of $\tau$ manifest the signal information contained in the datasets. Intuitively, singular set $\{\tau_1\}$ indicates the primary signal frequency, while sets $\{\tau_1, \tau_2\}$ and $\{\tau_1, \tau_2, \tau_3\}$ are able to uncover the signal distribution. Specifically, the fact that $\tau$ set $\{0.5, 0.8, 1.1\}$ is more effective than $\{0.8, 1.0\}$ on Cora manifests that high-frequency (larger $\tau$) signal benefits node representations. For the case of dataset Actor (resp. Chameleon), it hints that the high-frequency (resp. low-frequency) signal dominates the others.

**Orthogonal bases and computation overheads.** We exploit the three-term recurrence method to orthogonalize polynomial bases and implement the orthogonal version OrthKry of AdaptKry. Since OrthKry fixes the relative positions of polynomial bases into orthogonality, it cannot take advantage of the utilization of multiple bases. For a fair comparison, we adopt a singular basis for AdaptKry as well. In particular, we test OrthKry and AdaptKry on two homophily datasets Cora and Citeseer, and two heterophily datasets Chameleon and Squirrel.

Table 5 reports the accuracy scores with propagation hop $K \in \{2, 4, 6, 8, 10\}$. On the two homophily datasets Cora and Citeseer,

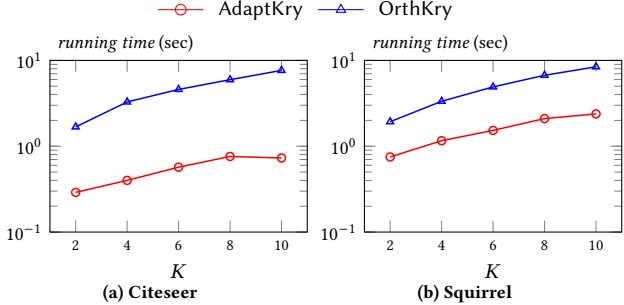

**Figure 4: Running times of orthgonal / non-orthogonal polynomial bases.**

AdaptKry achieves consistently higher accuracy scores than OrthKry does for different propagation hop $K$ values. On the heterophily dataset Chameleon, the performance of OrthKry closely aligns with that of AdaptKry as they outperform each other alternatively. On dataset Squirrel, we observe that OrthKry demonstrates performance gains over AdaptKry. In a nutshell, AdaptKry exhibits clear advantages over OrthKry on homophily datasets while none of them presents dominant filtering capability on heterophily datasets. Therefore, non-orthogonal polynomials in AdaptKry exhibit general and comparable filter capability with orthogonal polynomials.

Figure 4 compares the running times, i.e., feature propagation times and training times of AdaptKry and OrthKry on datasets Citeseer and Squirrel. As displayed, AdaptKry demonstrates notable efficiency advantages over OrthKry. The primary reason is that the adaptive Krylov basis in AdaptKry accommodates the underlying better than the fixed orthogonal basis in OrthKry, as proved in Theorem 3. As we also mentioned in Section 4.3, the orthogonal basis in OrthKry incurs extra $O(Kn)$ computation overheads compared with the non-orthogonal basis.

## 6 RELATED WORK

**Krylov subspace method.** The idea of the Krylov subspace method was originally conceived by Lanczos, Hestenes, and Stiefel in the early 1950s for solving linear algebraic systems [18, 32]. An order-$K$ Krylov subspace is constructed by multiplying the first $K$ powers of a $n \times n$ matrix $\mathbf{A}$ to an $n$-dimension vector $\mathbf{v}$, i.e., $\mathcal{K}_K(\mathbf{A}, \mathbf{v}) = \text{span}\{\mathbf{v}, \mathbf{A}\mathbf{v}, \cdots, \mathbf{A}^{K-1}\mathbf{v}\}$. In doing so, the Krylov subspace as a subspace of $\mathbb{R}^n$ can capture the property of $\mathbf{A}$ [36]. Specifically, for a large linear system $\mathbf{A}\mathbf{x} = \mathbf{v}$ to identify the unknown $\mathbf{x} \in \mathbb{R}^n$, the Krylov subspace method constructs a subspace $\mathcal{K}_m(\mathbf{A}, \mathbf{v})$ with $m \ll n$ and then finds an approximate solution $\tilde{\mathbf{x}}$ that belongs to $\mathcal{K}_m(\mathbf{A}, \mathbf{v})$. Krylov subspace method is widely used in iterative methods in linear algebra [29].

**Polynomial graph filters.** The aim of research on graph spectral filters is to enhance expressive capabilities through various polynomials to approximate the optimal graph filters. ChebNet [10] utilizes a truncated Chebyshev polynomial [19, 30] up to $K$ orders with each polynomial basis assigned a learnable parameter, which makes ChebNet a general localized graph filters [41]. By simplifying ChebNet, GCN [23] retains the first two convolution layers and takes a renormalized propagation matrix as convolution operations,

which makes it a low-pass filter. To better capture node proximity, APPNP [24] adopts Personalized PageRank (PPR) [33, 38] and takes the PPR values of neighbors as fixed aggregation weights, resulting in a low-pass graph filter as well. To achieve better adaptability, GPR-GNN [8] exploits the generalized PageRank (GPR) matrix with trainable weights for convolutional layers. Similarly, GNN-LF/HF proposed by Zhu et al. [53] forms a low-pass and high-pass filter respectively by devising the variants of the Laplacian matrix as the propagation matrices. ARMA [3] resorts to rational convolutional filters using auto-regression moving average filters [31] instead of polynomials, aiming to yield flexible frequency response. However, to approximate the inverse matrix, ARMA inevitably incurs large computation overheads. He et al. [21] employ Bernstein polynomials [12] and propose BernNet for better controllability and interpretability. However, feature propagation in BernNet leads to quadratic time complexity of hop $K$. To improve the flexible controllability on concentration center and bandwidth, Li et al. [28] combines a series of Gaussian bases on approximation and propose $G^2CN$. In the meanwhile, Wang and Zhang [44] investigate the expressive power of polynomial filters and propose JacobiConv by adopting Jacobi polynomial [1] bases. Later, He et al. [20] revisit Chebyshev approximation and point out the over-fitting issue in ChebNet. To resolve the issue, they propose ChebNetII based on Chebyshev interpolation. Recently, Guo and Wei [17] orthogonalize the polynomial bases for optimal convergence and propose OptBasisGNN. ASGC [6] simplifies graph convolution operation inspired by SGC [47] for heterophily graphs. To this end, a Krylov matrix is calculated and equipped with coefficients. However, ASGC presents suboptimal classification performance in the experiments. SIGN is another SGC-based method similar to AdaptKry. However, SIGN filters out the negative values in Krylov basis before weight learning. This operation damages the polynomial properties of the Krylov basis and unable to approximate graph filters. There are also other newly proposed GNN models, e.g., Nodeformer [49], NIGCN [22], and Auto-HeG [52], but they do not exhibit explicit spectral properties and thus are not discussed here.

## 7 CONCLUSION

In this paper, we unify existing polynomial filters and optimal filters into the Krylov subspace and reveal their limited adaptability for diverse heterophily graphs. To optimize polynomial filters, we design a novel adaptive Krylov subspace with provable controllability over the graph spectrum, enabling adaptability to graphs in varying heterophily degrees. Consequently, we propose AdaptKry by leveraging this adaptive Krylov basis. Furthermore, we extend AdaptKry to incorporate multiple adaptive Krylov bases without incurring extra training costs, thereby accommodating graphs with diverse spectral properties and offering insights into the inherent complexities. The extensive experiments and ablation studies strongly support the superior performance of AdaptKry as polynomial filters, as well as the optimized expressive capability of the adaptive Krylov basis.

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

# A APPENDIX

## A.1 Proofs

PROOF OF PROPOSITION 1. Suppose a $K$-order polynomial graph filter $GF$ with propagation matrix $\mathbf{P}_M$. Let $f(x, i) = \sum_{k=0}^{K} \phi_{ik} x^k$ be the $i$-th polynomial basis of $GF$ where $\Phi \in \mathbb{R}^{(K+1) \times (K+1)}$ is the coefficient matrix and $\Phi[i, k] = \phi_{ik}$. Consequently, representation $\mathbf{Z}$ from $GF$ is calculated as $\mathbf{Z} = \sum_{i=0}^{K} w_i f(\mathbf{P}_M, i) \mathbf{X} = \sum_{i=0}^{K} w_i \sum_{k=0}^{K} \phi_{ik} \mathbf{P}_M^k \mathbf{X}$ where $\mathbf{w} \in \mathbb{R}^{K+1}$ is the learnable parameter vector. By choosing $\Theta = \{\theta_0, \theta_1, \ldots, \theta_K\} \in \mathbb{R}^{K+1}$ with $\Theta = \Phi^\top \mathbf{w}$ as the coefficients of basis $(\mathbf{X}, \mathbf{P}_M \mathbf{X}, \mathbf{P}_M^2 \mathbf{X}, \ldots, \mathbf{P}_M^K \mathbf{X})$, we have $\sum_{i=0}^{K} \theta_i \mathbf{P}_M^i \mathbf{X} = \sum_{i=0}^{K} w_i \sum_{k=0}^{K} \phi_{ik} \mathbf{P}_M^k \mathbf{X}$, which completes the proof. □

PROOF OF PROPOSITION 2. Let $\{\mathbf{u}_i \mid i \in [n]\}$ be the orthogonal eigenvectors of $\mathcal{L}$, where $\mathbf{u}_i$ is associated with eigenvalue $\lambda_i$. Then $\mathbf{U} g_\mathbf{w}(\Lambda) \mathbf{U}^\top \mathbf{x} = \sum_{i=1}^{n} g_\mathbf{w}(\lambda_i) \mathbf{u}_i \mathbf{u}_i^\top \mathbf{x}$. Moreover, since $\mathbf{P}$ is the similar matrix of $\mathcal{L}$, so $\mathbf{P}$ owns the same eigenvectors of $\mathcal{L}$. Let $\mathcal{T}$ be a subset of $[n]$ with $|\mathcal{T}| = t$ that contains the index of $i \in [n]$ such that $u_i^\top \mathbf{x} \neq 0 \; \forall i \in \mathcal{T}$ and $u_j^\top \mathbf{x} = 0 \; \forall j \in [n] \setminus \mathcal{T}$. Then, we have $\sum_{k=0}^{t-1} \theta_k \mathbf{P}^k \mathbf{x} = \sum_{k=0}^{t-1} \sum_{i \in \mathcal{T}} \theta_k \lambda_i^k \mathbf{u}_i \mathbf{u}_i^\top \mathbf{x} = \sum_{i \in \mathcal{T}} (\sum_{k=0}^{t-1} \theta_k \lambda_i^k) \mathbf{u}_i \mathbf{u}_i^\top \mathbf{x}$. Therefore, $\sum_{k=0}^{t-1} \theta_k \lambda_i^k = g_\mathbf{w}(\lambda_i)$ holds with proper $\theta_k$ for $i \in \mathcal{T}$. □

PROOF OF THEOREM 1. The propagation matrix $\mathbf{P}$ can be decomposed as $\mathbf{P} = \mathbf{U} \Lambda \mathbf{U}^{-1}$ where the $i$-th column of $\mathbf{U}$ is the eigenvector of $\lambda_i(\mathbf{P})$, denoted as $\phi_i$. Then we have

$$\lim_{K \to \infty} \mathbf{P}^K = \lim_{K \to \infty} (\Phi \Lambda \Phi^{-1})^K$$
$$= \lim_{K \to \infty} \Phi \Lambda^K \Phi^{-1}$$
$$= \Phi \lim_{K \to \infty} \Lambda^K \Phi^{-1}$$
$$= \Phi \text{diag}[\lim_{K \to \infty} \lambda_1^K, \ldots, \lim_{K \to \infty} \lambda_n^K] \Phi^{-1}$$
$$= \Phi \text{diag}[0, 0, \ldots, 1] \Phi^{-1}$$

Thus we have $\lim_{K \to \infty} \mathbf{P}^K = \mathbf{P}_\pi = \phi_n \cdot \phi_n^\top$ where $\phi_n = \frac{\mathbf{D}^{\frac{1}{2}} \cdot \mathbf{1}}{\sqrt{2m}} \in \mathbb{R}^n$. Therefore, we have $\mathbf{P}_\pi[u, v] = \frac{\sqrt{d_u d_v}}{2m}$.

Let $\mathbf{e}_u \in \mathbb{R}^{1 \times n}$ be an indicator vector having 1 in coordinate $u \in \mathcal{V}$. Then $\mathbf{P}^K[u, v]$ is expressed as $\mathbf{P}^K[u, v] = \mathbf{e}_u \mathbf{P}^K \mathbf{e}_v^\top$. For $\mathbf{e}_u$ and $\mathbf{e}_v$, we decompose $\mathbf{e}_u = \sum_{i=1}^{n} \alpha_i \phi_i^\top$, $\mathbf{e}_v = \sum_{i=1}^{n} \beta_i \phi_i^\top$. Notice that $\alpha_n = \frac{\sqrt{d_u}}{\sqrt{2m}}$ and $\beta_n = \frac{\sqrt{d_v}}{\sqrt{2m}}$. We have

$$\max_{u, v \in \mathcal{V}} \frac{|\mathbf{P}^K[u, v] - \mathbf{P}_\pi[u, v]|}{\mathbf{P}_\pi[u, v]}$$
$$= \max_{u, v \in \mathcal{V}} \frac{|\mathbf{e}_u \mathbf{P}^K \mathbf{e}_v^\top - \mathbf{P}_\pi[u, v]|}{\mathbf{P}_\pi[u, v]}$$
$$\leq \max_{u, v \in \mathcal{V}} \frac{\sum_{i=1}^{n-1} |\lambda_i(\mathbf{P})^K \alpha_i \beta_i|}{\mathbf{P}_\pi[u, v]}$$
$$\leq \lambda^K \cdot \max_{u, v \in \mathcal{V}} \frac{\sum_{i=1}^{n-1} |\alpha_i \beta_i|}{\mathbf{P}_\pi[u, v]}$$
$$\leq \lambda^K \cdot \max_{u, v \in \mathcal{V}} \frac{\|e_u\| \|e_v\| \cdot 2m}{\sqrt{d_u d_v}}$$
$$\leq \lambda^K \cdot \frac{2m}{d_{\min}}$$

where $\lambda = \max\{-\lambda_1(\mathbf{P}), \lambda_{n-1}(\mathbf{P})\}$ and $d_{\min} = \min\{d_v : v \in V\}$. Therefore, if $K \geq \ln \frac{\epsilon d_{\min}}{2m} / \ln \lambda$, $\lambda^K \cdot \frac{2m}{d_{\min}} \leq \epsilon$, which completes the proof. □

PROOF OF THEOREM 2. W.l.o.g, we set $t = \frac{1-\tau}{\tau}$ for $\tau > 0$. In particular, $t \in (-1, 0)$ for $\tau > 1$ and $t \in [0, \infty)$ for $\tau \in (0, 1]$. Thus, we have $\mathbf{P}_\tau = (\mathbf{D} + t\mathbf{I})^{-\frac{1}{2}} (\mathbf{A} + t\mathbf{I}) (\mathbf{D} + t\mathbf{I})^{-\frac{1}{2}}$. Assume that $\mathbf{v}_1, \ldots, \mathbf{v}_{i-1}$ are eigenvectors of $\mathcal{L}_\tau$ corresponding to eigenvalues $\lambda_1(\tau), \ldots, \lambda_{i-1}(\tau)$, respectively. According to Courant–Fischer theorem [13], we have

$$\lambda_i(\tau) = \min_{\mathbf{x}: \mathbf{v}_j^\top \mathbf{x} = 0 \text{ for } j \in [i-1]} \frac{\mathbf{x}^\top \mathcal{L}_\tau \mathbf{x}}{\mathbf{x}^\top \mathbf{x}}$$
$$= \min_{\mathbf{x}: \mathbf{v}_j^\top \mathbf{x} = 0 \text{ for } j \in [i-1]} \frac{\mathbf{x}^\top (\mathbf{D} + t\mathbf{I})^{\frac{1}{2}} \mathcal{L}_\tau (\mathbf{D} + t\mathbf{I})^{\frac{1}{2}} \mathbf{x}}{\mathbf{x}^\top (\mathbf{D} + t\mathbf{I}) \mathbf{x}}$$
$$= \min_{\mathbf{x}: \mathbf{v}_j^\top \mathbf{x} = 0 \text{ for } j \in [i-1]} \frac{\sum_{\langle u, v \rangle \in \mathcal{E} \cup \{\langle u, v \rangle | \forall u \in V\}} (\mathbf{x}_u - \mathbf{x}_v)^2}{\sum_{u \in \mathrm{v}} (d_u + t) \mathbf{x}_u^2}$$
$$= \min_{\mathbf{x}: \mathbf{v}_j^\top \mathbf{x} = 0 \text{ for } j \in [i-1]} \frac{\sum_{\langle u, v \rangle \in \mathcal{E}} (\mathbf{x}_u - \mathbf{x}_v)^2}{\sum_{u \in \mathrm{v}} (d_u + t) \mathbf{x}_u^2}$$

Notice that when $\tau$ increases, $t$ decreases. Consequently, $\lambda_i(\tau)$ increase. Meanwhile, we also get $\lambda_i(\tau) \leq \lambda_i(1) = \lambda_i$ for $\tau \in (0, 1]$ and $\lambda_i(\tau) > \lambda_i(1)$ for $\tau > 1$, which completes the proof. □

PROOF OF THEOREM 3. Let $\mathbf{y} \in \mathbb{N}^n$ be the label vector, i.e., $\mathbf{Y}_u = i$ when node $u$ belongs to the $i$-th class. Let $\mathcal{L} = \mathbf{U} \Lambda \mathbf{U}^\top$ where $\mathbf{U}$ is the eigenvector matrix and $\Lambda$ is the diagonal matrix of eigenvalue spectrum $\{\lambda_1, \lambda_2, \cdots, \lambda_n\}$. Suppose $\mathbf{U}^\top \mathbf{y} = (\alpha_1, \alpha_2, \cdots, \alpha_n)$ where $\alpha_i$ is the projection (response) of $\mathbf{y}$ on eigenvector $\mathbf{U}^\top[i, :]$. Then we have $\mathbf{y}^\top \mathcal{L} \mathbf{y} = \mathbf{y}^\top \mathbf{U} \Lambda \mathbf{U}^\top \mathbf{y} = \sum_{i=1}^{n} \alpha_i^2 \lambda_i$.

W.l.o.g, we have $\mathbf{F}_1(\mathbf{x}) = \sum_{k=0}^{t-1} \mathbf{w}_k^o \mathbf{P}^k \mathbf{x} = \sum_{k=0}^{t-1} \sum_{i=1}^{n} \mathbf{w}_k^o \lambda_i^k \psi_i \psi_i^\top \mathbf{x}$ and $\mathbf{F}_2(\mathbf{x}) = \sum_{k=0}^{t-1} \mathbf{w}_k^a \mathbf{P}_\tau^k \mathbf{x} = \sum_{k=0}^{t-1} \sum_{i=1}^{n} \mathbf{w}_k^a \lambda_i^k(\tau) \phi_i \phi_i^\top \mathbf{x}$ where $\lambda_i$ (resp. $\lambda_i(\tau)$) is the $i$-th eigenvalue and $\psi_i$ (resp. $\phi_i$) is the associated eigenvector of $\mathbf{P}$ (resp. $\mathbf{P}_\tau$). For ease of exposition, we assume $\psi_i^\top \mathbf{x} = \gamma_i$ and $\phi_i^\top \mathbf{x} = \beta_i$ for $i \in \{1, 2, \cdots, n\}$. Therefore, for the corresponding frequencies of $\mathbf{F}_1(\mathbf{x})$ and $\mathbf{F}_2(\mathbf{x})$, we have

$$\sum_{i=1}^{n} \sum_{k=0}^{t-1} \mathbf{w}_k^o \gamma_i^2 \lambda_i^k = \sum_{i=1}^{n} \sum_{k=0}^{t-1} \mathbf{w}_k^a \beta_i^2 \lambda_i^k(\tau) = \sum_{i=1}^{n} \alpha_i^2 \lambda_i \quad (10)$$

For better demonstration, we represent Equation (10) in the form of

$$(\gamma_1^2, \gamma_2^2, \cdots, \gamma_n^2)^\top \Gamma(\mathbf{w}_0^o, \mathbf{w}_1^o, \cdots, \mathbf{w}_{t-1}^o)$$
$$= (\beta_1^2, \beta_2^2, \cdots, \beta_n^2)^\top \Gamma_\tau(\mathbf{w}_0^a, \mathbf{w}_1^a, \cdots, \mathbf{w}_{t-1}^a)$$
$$= (\alpha_1^2, \alpha_2^2, \cdots, \alpha_n^2)^\top (\lambda_1, \lambda_2, \cdots, \lambda_n) \quad (11)$$

where $\Gamma_\tau$ (resp. $\Gamma$) $\in \mathbb{R}^{n \times t}$ is the Vandermonde matrix of $\lambda_i(\tau)$ (resp. $\lambda_i$). Specifically, $\Gamma_\tau = \begin{vmatrix} 1 & \lambda_1(\tau) & \lambda_1^2(\tau) & \cdots & \lambda_1^{t-1}(\tau) \\ 1 & \lambda_2(\tau) & \lambda_2^2(\tau) & \cdots & \lambda_2^{t-1}(\tau) \\ \cdots & \cdots & \cdots & \cdots & \cdots \\ 1 & \lambda_{n-1}(\tau) & \lambda_{n-1}^2(\tau) & \cdots & \lambda_{n-1}^{t-1}(\tau) \\ 1 & \lambda_n(\tau) & \lambda_n^2(\tau) & \cdots & \lambda_n^{t-1}(\tau) \end{vmatrix}$.

Note that $(\gamma_1^2, \gamma_2^2, \cdots, \gamma_n^2)$ and $\Gamma$ in $\mathbf{F}_1(\mathbf{x})$ are constant while $(\beta_1^2, \beta_2^2, \cdots, \beta_n^2)$ and $\Gamma_\tau$ in $\mathbf{F}_2(\mathbf{x})$ are functions of $\tau$. Meanwhile, as proved in Theorem 2, $\lambda_i(\tau)$ is a monotonically increasing

function of $\tau$ with controllable values, i.e., either $\lambda_i(\tau) \leq \lambda_i$ or $\lambda_i(\tau) > \lambda_i$. In particular, $\mathbf{F}_2(\mathbf{x})$ is a general extension of $\mathbf{F}_1(\mathbf{x})$, e.g., $(\beta_1^2, \beta_2^2, \cdots, \beta_n^2) = (\gamma_1^2, \gamma_2^2, \cdots, \gamma_n^2)$ and $\Gamma_\tau = \Gamma$ when $\tau = 1$.

As a consequence, for a randomly initialized weight $\mathbf{w}$, let $\{\mathbf{w}, \mathbf{w}_1, \mathbf{w}_2, \cdots, \mathbf{w}_p, \mathbf{w}^o\}$ be the learning trace of a model from $\mathcal{K}_t(\mathbf{P}, \mathbf{x})$ to optimal filter $\mathbf{F}_1$ after training in $(p+1)$ steps. In thus a case, there exists an appropriate $\tau > 0$ to alter $\gamma_i$ into $\beta_i$ and reshape $\Gamma$ into $\Gamma_\tau$ to reach $\mathbf{w}^a$ before $(p+1)$ steps, ensuring Equation (11). Put differently, obtaining $\mathbf{w}^a$ is more attainable than reaching $\mathbf{w}^o$ starting from the initial weight $\mathbf{w}$ in expectation, i.e., $\mathbb{E}[\|\mathbf{w}^a - \mathbf{w}\|_2] \leq \mathbb{E}[\|\mathbf{w}^o - \mathbf{w}\|_2]$. □

**Table 6: Hyperparameters of AdaptKry.**

| Datasets | Hop $K$ | $\tau$ | Learning rate | Hidden dimension |
|---|---|---|---|---|
| Cora | 10 | {0.5,0.8,1.1} | 0.10 | 256 |
| Citeseer | 10 | 0.1 | 0.01 | 128 |
| Pubmed | 10 | 0.5 | 0.10 | 128 |
| Actor | 10 | { 0.6,1.7,1.8} | 0.10 | 256 |
| Chameleon | 10 | { 0.5,0.8} | 0.01 | 256 |
| Squirrel | 10 | 0.8 | 0.005 | 256 |

## A.2 Experimental Settings

**Running Environment.** All experiments are conducted on a Linux machine with an NVIDIA Tesla V100 (32GB memory), Intel Xeon(R) CPU (2.80GHz), and 500GB RAM.

**Parameter Settings.** Table 6 presents the hyperparamters (hop $K$, $\tau$, learning rate and hidden dimension) of AdaptKry adopted on the 6 small datasets and 4 large datasets. As said in Section 5.1, we fix $K = 10$ for the small datasets and tune $K$ in $\{2, 4, 6, 8, 10\}$ for the large datasets by following the setting in [20]. For $\tau$, we tune $\tau$ in the range $[0.1, 1]$. In particular, we normally set $\tau = 0.9$ for the three strong heterophily graphs, as verified in our experiment (Section 5.3, and tune it for the rest datasets for the best fit. Learning rate (lr) is selected from the set $\{0.005, 0.01, 0.05, 0.10\}$ and hidden dimension is tuned from the set $\{128, 256, 512, 1024, 2048\}$. For baselines, we either adopt their recommended parameter settings on corresponding datasets or tune their parameters carefully following the above scheme for a fair comparison.

**Implementation Details.** We implement AdaptKry in PyTorch. The baselines are obtained from their official release. Table 7 summarizes the links to baseline codes. As with ChebNet and GPR-GNN, we adopt their implementations inside the source code of BernNet.

**Table 7: Download links of baseline methods.**

| Methods | URL |
|---|---|
| GCN | https://github.com/pyg-team/pytorch_geometric |
| SGC | https://github.com/Tiiiger/SGC |
| ASGC | https://github.com/schariya/adaptive-simple-convolution |
| SIGN | https://github.com/twitter-research/sign |
| BernNet | https://github.com/ivam-he/BernNet |
| JacobiConv | https://github.com/GraphPKU/JacobiConv |
| EvenNet | https://github.com/leirunlin/evennet |
| ChebNetII | https://github.com/ivam-he/ChebNetII |
| OptBasisGNN | https://github.com/yuziGuo/FarOptBasis |

**Table 8: Accuracy (%) for orthogonal / non-orthogonal polynomials**

| Dataset | Method | hop $K$ | | | | |
|---|---|---|---|---|---|---|
| | | 2 | 4 | 6 | 8 | 10 |
| Pubmed | OrthKry | 91.56 | 91.72 | 91.47 | 91.68 | 91.47 |
| | AdaptKry | 92.17 | 91.90 | 91.95 | 92.01 | 92.05 |
| Actor | OrthKry | 40.51 | 40.71 | 40.33 | 40.04 | 39.11 |
| | AdaptKry | 39.87 | 40.92 | 41.21 | 41.92 | 42.06 |

## A.3 Additional Experimental Results

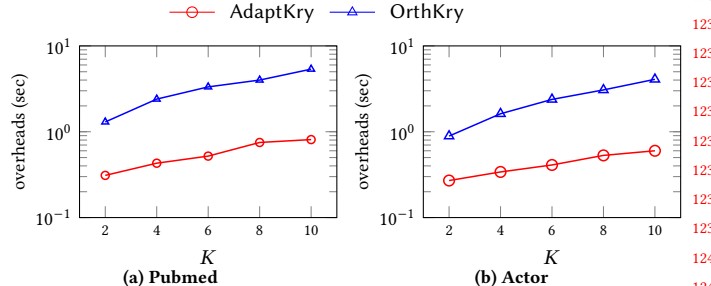

**Figure 5: Non-orthgonal and orthogonal polynomial computation overheads.**

**Further exploration on orthogonality.** For a complete evaluation, we further test AdaptKry and OrthKry on datasets Pubmed and Actor to compare non-orthogonal and orthogonal polynomials in graph filters. Table 8 presents the accuracy scores on the homophily dataset Pubmed and the heterophily dataset Actor. We observe that AdaptKry shows a clear advantage over OrthKry on Pubmed. Except for the case of $K = 2$, AdaptKry also beats OrthKry on heterophily dataset Actor. Together with the results in Table 5, neither AdaptKry nor OrthKry graph filter demonstrates dominance over the other across various graphs, and our experiments indicate that orthogonal polynomials do not exhibit clear advantages over non-orthogonal polynomials.

Meanwhile, we also plot the polynomial computation overheads of AdaptKry and OrthKry on datasets Pubmed and Actor in Figure 5. Similar to the results in Figure 4, the computation overheads of non-orthogonal polynomials in AdaptKry can be one order of magnitude smaller than those of orthogonal polynomials in OrthKry. Those findings substantiate the superior efficacy of non-orthogonal bases in comparison to orthogonal bases.

**Learned graph filters.** To better illustrate the graph filtering capability of AdaptKry, we plot the graph filters learned on the 3 homophily graphs (Cora, Citeseer, Pubmed) and 3 heterophily graphs (Actor, Chameleon, Squirrel) in Figure 6. As seen, AdaptKry learns explicit low-pass filters on homophily graphs and a high-pass filter on Actor, comb-pass filters on Chameleon and Squirrel, coherent with the results in the literature [21].

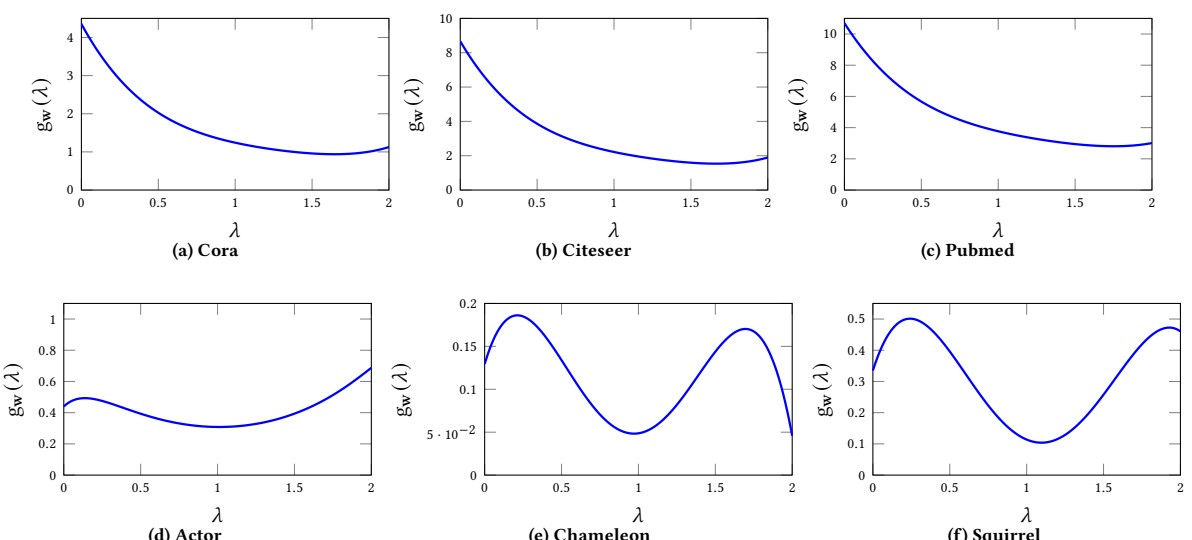

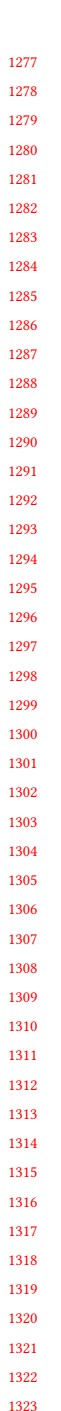

**Figure 6: Graph filters learnt by AdaptKry on 6 graphs with various homophily ratios**

