# OpenReview forum: "Optimizing Polynomial Graph Filters: A Novel Adaptive Krylov Subspace Approach"
_ACM.org/TheWebConf/2024/Conference — TheWebConf24_

### Official Review · Reviewer_qLZn · 2023-11-21

**Novelty:** 5
**Technical Quality:** 5

**Review:**

**Summary**
The paper introduces AdaptKry, a novel approach for optimizing polynomial graph filters. It unifies different filters into a Krylov subspace, improves adaptability to varying graph structures, and extends to handle complex graphs efficiently. Experimental results show AdaptKry's superior filtering capabilities and its ability to capture intricate graph characteristics.

**Strength**
1. Unifying polynomial filters into the Krylov subspace is an interesting idea.
2. AdaptKry performs well on some datasets.
3. The paper is well-written and easy to read.

**Weakness**
1. Proposition 2 defines the optimal polynomial filters, and it seems that AdaptKry does not satisfy this Proposition. I would like to understand how much AdaptKry differs from the optimal polynomial filters.
2. The main differences between AdaptKry and methods based on polynomial filters lie in the use of a propagation matrix with tao and the final concatenation operation. Because the concatenation operation increases more parameters, I believe it is necessary to conduct ablation experiments to demonstrate that the benefits of AdaptKry do not solely come from the concatenation operation.
3. The improvement of AdaptKry on heterophilic graphs is marginal.
4. There are not enough datasets for heterophilic graphs. Please note that previous work has highlighted some issues with the Chameleon and Squirrel datasets [1]. Therefore, I recommend conducting experiments using more extensive heterophilic graph datasets, such as the latest benchmarks available [1] and [2].

[1] Platonov, Oleg, et al. "A critical look at the evaluation of GNNs under heterophily: Are we really making progress?." _The Eleventh International Conference on Learning Representations_. 2022.

[2] Lim, Derek, et al. "Large scale learning on non-homophilous graphs: New benchmarks and strong simple methods." _Advances in Neural Information Processing Systems_ 34 (2021): 20887-20902.

**Questions:**

Please refer to the Weaknesses as described above for details.

**Reviewer Confidence:**

4: The reviewer is certain that the evaluation is correct and very familiar with the relevant literature

**Scope:**

4: The work is relevant to the Web and to the track, and is of broad interest to the community

---

### Official Review · Reviewer_LgWU · 2023-11-21

**Novelty:** 4
**Technical Quality:** 4

**Review:**

**Summary**
This work proposes a differentiable framework for (undirected and unweighted)
graph filters that generalizes several well-studied polynomial approximations:
ChebNet, GPR-GNN, BernNet, and JacobiConv. In particular, it proposes using
Krylov subspaces (based on the normalized Laplacian matrix with a parameterized
amount of self-loops) and trainable per-term scale factors. The `AdaptKry`
algorithm then feeds this projection into an MLP as input for the downstream
learning task. The paper also provides extensive experiments for node
classification on academic benchmarks.

**Strengths**
- The Krylov subspace approach with $K$ trainable parameters generalizes several
  previous works on (spectral) graph filters with polynomial approximations:
  * ChebNet (Defferrard et al., NIP 2016)
  * GPR-GNN (Chien et al., ICLR 2021)
  * BernNet (He et al., NeurIPS 2021)
  * JacobiConv (Wang-Zhang, ICML 2022)
- The paper is well organized and mostly well written.
- The node classification experiments are comprehensive and compare to many
  previous works.

**Weaknesses**
- Theorem 1 is fairly well known. Please cite this result, e.g., from the
  Markov chain mixing time literature.
- Section 3.3 could benefit from being made more rigorous, e.g., quantifying
  how much information could we lose if $K$ is too small.

**Typos and suggestions**
- [line 67] nit: "a graph $G$" --> "an undirected graph $G$" since you're using
  its eigendecomposition.
- [line 201] suggestion: "For simplicity purposes," --> "For simplicity,"
- [line 206] typo: "Normalized Laplacian matrix" --> "The normalized Laplacian
  matrix"
- [line 215] suggestion: Remove "Without loss of generality," if this is always
  true (i.e., without needing to assume anything).
- [line 240] typo: "ration" --> "ratio"
- [line 242] suggestion: consider using $(u,v)$ or $\{u,v\}$ for edges since
  $\langle u, v \rangle$ looks like an inner product without context.

**Questions:**

- [line 329] Isn't the grade of $(x, P)$ just the rank of the Krylov subspace?
  If so, why introduce new notation?
- [line 356] Up until this point in the paper, there was no mention of graphs
  not being bipartite. If this assumption is needed for your results, it makes
  sense to make the assumption more explicit early on in Section 2.

**Reviewer Confidence:**

3: The reviewer is confident but not certain that the evaluation is correct

**Scope:**

4: The work is relevant to the Web and to the track, and is of broad interest to the community

---

### Official Review · Reviewer_SQaZ · 2023-11-24

**Novelty:** 5
**Technical Quality:** 5

**Review:**

This paper unifies the polynomial filters through the Krylov subspace perspective. And authors demonstrate that polynomial filters included in the Krylov subspace of the same order has an equivalent expressive power theoretically. The authors propose AdaptKry to learn the graph spectrum filters, and utilize  graph heat equation to design propagation matrix $\mathbf{P}$ to construct the Krylov subspace.
In experiments, AdaptKry shows best scores in most real world datasets in node classification task. Ablation study demonstrates the influence of parameter $\tau$ on different datasets.

**Strengths:**

1. Unifying poly-based spectral GNNs via Krylov subspace perspective is inspiring.
2. The introduced model can make a superior performance on most datasets.
3. Writing and expression are clear.

**Weaknesses:**

1. Could you provide further insights regarding AdaptKry's potential to learn more complex filters, such as comb or band-pass filters? I am uncertain whether Figure 1 represents the results obtained from AdaptKry.
2. Avoiding eigendecomposition is a significant advantage when utilizing polynomials as filter bases. However, it seems that the experimental section could be further strengthened by evaluating on large-scale graphs.
3. The selection of propagation matrix $\mathbf{P}_\tau$ seems like parameterization normalized Laplacian matrix.  Can i say the performance improvment most depends on the specific propagation matrix?  It should be more disccusion about the propagation matrix.

**Questions:**

see weakness.

**Reviewer Confidence:**

4: The reviewer is certain that the evaluation is correct and very familiar with the relevant literature

**Scope:**

4: The work is relevant to the Web and to the track, and is of broad interest to the community

---

### Official Review · Reviewer_w9rP · 2023-11-27

**Novelty:** 5
**Technical Quality:** 5

**Review:**

This paper theoretically unifies polynomial graph filters and optimal filters of identical degrees into the Krylov subspace, offering equivalent expressive power. Further exploration of the asymptotic convergence properties of polynomials from the unified Krylov subspace perspective reveals their limited adaptability in heterogeneous graphs. Inspired by these findings, an adaptive Krylov subspace approach AdaptKry is proposed to optimize polynomial bases and can be extended by multiple adaptive Krylov bases for complex graphs. The proposed AdaptKry demonstrates superior filtering capabilities and optimized efficacy of the adaptive Krylov basis on real-world datasets.

**Questions:**

1. Based on Equation 6, polynomial filters utilize various propagation matrices P. Here, I have some confusion: (a) Why is the propagation matrix P for BernNet I-L/2 and I-L for JacobiConv at Line 353-354? (b) Are the propagation matrices in Equations 4 and 5 linearly correlated with the normalized Laplacian matrix L?

2. Can you theoretically or experimentally demonstrate the relationship between tau and the homophily ratio? Providing some evidence for selecting tau based on the homophily ratio would be valuable, and you may consider incorporating an ablation study to further investigate this.

3. At line 3 of Algorithm 1, is there a computational cost or numerical stability issue associated with the matrix inverse operation of the normalized adjacency matrix? Have any optimizations from PyTorch Geometric been utilized? Could you compare the runtime with baselines, especially on large graphs?

4. The performance on larger datasets should be verified like BernNet and ChebNetII, such as the arXiv dataset. In the section regarding parameter settings at line 1190, only 6 datasets are mentioned, not 10.

**Reviewer Confidence:**

3: The reviewer is confident but not certain that the evaluation is correct

**Scope:**

4: The work is relevant to the Web and to the track, and is of broad interest to the community

---

### Official Review · Reviewer_Wxf6 · 2023-11-28

**Novelty:** 5
**Technical Quality:** 5

**Review:**

Summary:

This paper considers the design of graph spectral filters via consideration of Krylov subspaces generated by input feature vectors $x$ and propagation matrices $P$.  It notes that for a fixed signal $x$ and propagation matrix $P$, the vector space generated by $x, Px, ..., P^k x, ...$ is finite-dimensional (say, dimension $t$), and so the spectral filter given by diagonalization of the graph Laplacian is equivalent to some polynomial filter of appropriate degree.  It then connects the homophily ratio of a signal on a graph $G$ to its frequency responses.  Motivated by this, it proposes an adaptive spectral filter with a trainable parameter.  The paper then proves that this parameter can be varied to accommodate signals with a wide range of homophily ratios.

Pros:

1.) The paper gives solid theoretical motivation for its approach.

2.) The problem of the design of spectral filters for heterophilous graph signals is an important one.

3.) The empirical results seem promising.

Cons:

1.) The writing is confusing at times.  Throughout, there is some awkward wording.

2.) I am not sure that Theorem 1 is particularly novel.  It seems to be some variant of a theorem giving a convergence guarantee for the power method (for approximating the largest eigenpair of a matrix).

**Questions:**

1.) Can the authors respond to my second con point?

**Reviewer Confidence:**

3: The reviewer is confident but not certain that the evaluation is correct

**Scope:**

4: The work is relevant to the Web and to the track, and is of broad interest to the community

---

### Decision · Program_Chairs · 2024-01-22

**Decision:**

Accept

**Comment:**

The reviewers are in agreement about the result's high relevance, technical quality, and novelty. There are, however, sufficient concerns about technical details, as well as experimental comaprisons/scope, that make this result feel like an intermediate step to me.